# Tunable room-temperature ferromagnetism in Co-doped two-dimensional van der Waals ZnO

Rui Chen[1,2], Fuchuan Luo[1,3], Yuzi Liu [4], Yu Song[2,5], Yu Dong[6], Shan Wu[2,5], Jinhua Cao[1,2], Fuyi Yang[1,2], Alpha N'Diaye [7], Padraic Shafer [7], Yin Liu [1,2], Shuai Lou[1,2], Junwei Huang [6], Xiang Chen [2,5], Zixuan Fang [1,3], Qingjun Wang[1,2], Dafei Jin [4], Ran Cheng [8], Hongtao Yuan[6], Robert J. Birgeneau [1,2,5] & Jie Yao [1,2 ✉]

The recent discovery of ferromagnetism in two-dimensional van der Waals crystals has provoked a surge of interest in the exploration of fundamental spin interaction in reduced dimensions. However, existing material candidates have several limitations, notably lacking intrinsic room-temperature ferromagnetic order and air stability. Here, motivated by the anomalously high Curie temperature observed in bulk diluted magnetic oxides, we demonstrate room-temperature ferromagnetism in Co-doped graphene-like Zinc Oxide, a chemically stable layered material in air, down to single atom thickness. Through the magneto-optic Kerr effect, superconducting quantum interference device and X-ray magnetic circular dichroism measurements, we observe clear evidences of spontaneous magnetization in such exotic material systems at room temperature and above. Transmission electron microscopy and atomic force microscopy results explicitly exclude the existence of metallic Co or cobalt oxides clusters. X-ray characterizations reveal that the substitutional Co atoms form $Co^{2+}$ states in the graphitic lattice of ZnO. By varying the Co doping level, we observe transitions between paramagnetic, ferromagnetic and less ordered phases due to the interplay between impurity-band-exchange and super-exchange interactions. Our discovery opens another path to 2D ferromagnetism at room temperature with the advantage of exceptional tunability and robustness.

[1] Department of Materials Science and Engineering, University of California, Berkeley, CA, USA. [2] Materials Sciences Division, Lawrence Berkeley National Laboratory, Berkeley, CA, USA. [3] National Engineering Research Center of Electromagnetic Radiation Control Materials, University of Electronic Science and Technology of China, Chengdu, P. R. China. [4] Center for Nanoscale Materials, Nanoscience and Technology Division, Argonne National Laboratory, Lemont, IL, USA. [5] Department of Physics, University of California, Berkeley, CA, USA. [6] National Laboratory of Solid-State Microstructures, College of Engineering and Applied Sciences, and Collaborative Innovation Center of Advanced Microstructures, Nanjing University, Nanjing, P. R. China. [7] Advanced Light Source, Lawrence Berkeley National Laboratory, Berkeley, CA, USA. [8] Department of Electrical and Computer Engineering, University of California, Riverside, CA, USA. ✉email: yaojie@berkeley.edu

Magnetic materials at their two-dimensional (2D) limit exhibit fundamentally new behaviors because of the deviation from perfect Heisenberg symmetry. The introduction of magnetic anisotropy and other anisotropic interactions can lift the Mermin–Wagner constraint on the long-range magnetic order. Ultra-thin van der Waals (vdW) materials, in particular, have provided us with excellent platforms for the exploration of 2D magnetism. Long-range ferromagnetic order has been successfully observed in 2D vdW crystals such as $Cr_2Ge_2Te_6$, $CrI_3$, and $Fe_nGeTe_2$ ($n = 3$, 4, and 5)[1–7]. In the presence of spontaneous magnetization, spin and charge degrees of freedom are further entangled and give rise to exceptional spintronic phenomena, such as giant tunneling magnetoresistance[8,9], magneto-electric coupling[10–12], and spin-orbit torque[13]. Although it is of great importance in fundamental physics and materials science to achieve ferromagnetic ground states in monolayers of the above materials, they still suffer from serious challenges including, but not limited to, low intrinsic Curie temperatures, air instability, as well as the complexity of magnetic property manipulations[14]. Also, previous characterizations are often limited by the ultra-low mass of atomically thin 2D magnets, which is intrinsically beyond the sensitivity of the superconducting quantum interference device (SQUID)[14]. Thus, a direct and in-depth investigation into 2D magnetic phase transitions is still lacking.

Diluted magnetic oxides (DMOs) may provide a new opportunity to realize 2D ferromagnetism and solve the above-mentioned issues. The past 20 years have witnessed unprecedented breakthroughs in three-dimensional (3D) DMOs, such as transition metal doped wurtzite ZnO[15–21]. Superior ferromagnetic performance has been uncovered in the DMO materials, which harbor high Curie temperatures beyond room temperature. The percolation model of bound magnetic polarons (BMP), which favors ferromagnetic coupling, has been widely discussed in such DMO systems in the limit of low carrier density or equivalently strong carrier localization[18,22], whose interplay with short-range antiferromagnetic superexchange interactions enables a new degree of freedom to tune the magnetic coupling by modulating the doping levels. Therefore, the magnetic behaviors can be effectively manipulated from paramagnetic to ferromagnetic, and eventually, to antiferromagnetic states.

Motivated by the abundant unique advantages of 3D DMOs, graphitic ZnO (gZnO) is regarded as an ideal model system to explore the spin couplings in 2D materials. gZnO is a layered wide-bandgap oxide material with a graphene-like honeycomb structure and strong air stability[23–30]. Theoretical calculations[23,26–28], surface X-ray diffraction[24], and transmission electron microscopy (TEM)[25,29,30] have revealed that ZnO is able to adopt a 2D vdW phase by removing the out-of-plane destabilizing dipoles when the sample thickness is smaller than three monolayers. gZnO has been predicted to be a promising host of DMOs owing to the barrierless incorporation of Co atoms and the remarkably strong entanglement between these substitutional Co ions[31].

Here we report the observation of room-temperature ferromagnetism in 2D graphitic $Zn_{1-x}Co_xO$ (gZCO) monolayer with strong environmental stability. Magnetism is successfully introduced to a non-magnetic 2D vdW crystal, gZnO, through the doping of Co atoms. Varying doping levels drives the transition from paramagnetic to ferromagnetic and then to a less-ordered state. In the following sections, we show our experimental observation of room-temperature 2D ferromagnetism in Co-doped gZnO. In stark contrast to previously reported 2D ferromagnets, the robustness of our system in ambient conditions greatly relaxes the stringent requirements in the further exploration of 2D magnetism at room temperature. In the meantime, contamination from the formation of clustered Co or cobalt oxides is completely ruled out, as evidenced by atomic force microscopy (AFM), TEM, X-ray techniques, and magneto-optical Kerr effect (MOKE) microscopy. Our approach will enable better understandings of exchange interactions in DMOs and provide a new platform for future 2D magnetic applications.

## Results

A graphene oxide (GO)-template-based approach[25] is exploited to synthesize 2D gZCO nanosheets (see Methods). It is worth noting that the final annealing process is conducted at 400 °C in air forone hour to remove the reduced GO (rGO) templates, which highlights the strong chemical stability of gZCO crystals in ambient conditions. Figure 1a schematically shows the crystal structure of monolayer gZCO. This honeycomb lattice of Zn and O atoms resembles that of hexagonal boron nitride. Substitutional Co dopants occupy Zn sites. The morphology of the gZCO nanosheet is verified by scanning electron microscopy (SEM, Fig. 1b) and AFM (Fig. 1c). AFM mapping also shows a monolayer with a thickness of 2.8 Å[25]. Notably, such an atomically smooth surface is free of nanoparticles, unambiguously suggesting the absence of magnetic contaminants, such as clustered Co, cobalt oxides, or extrinsic ones[32].

In order to gain comprehensive insights into the magnetic properties in the synthesized 2D gZCO, we first conduct longitudinal MOKE measurements to probe the local magnetization of individual gZCO single layer at 300 K. The MOKE result on a monolayer gZCO is shown in Fig. 1d. Room-temperature ferromagnetism is evidenced by a characteristic ferromagnetic hysteresis loop[1,2] with a coercive field of 400 Oe.

SQUID measurements are also performed. SQUID has long been regarded as the optimal technique for magnetic characterizations, whereas it is recently challenged when testing 2D atomic layers: owing to a significantly smaller mass load, real magnetic signals from atomically thin 2D sheets can be overwhelmed by the impurities in the substrates[14]; thus, 2D magnets cannot be probed with SQUID in a conventional manner. A notable advantage of our approach is that it allows characterizations based on composites composed of alternating gZCO and rGO layers (Fig. 1e, also see Methods), where ZCO is kept in its 2D vdW phase between adjacent rGO layers[25]. Meanwhile, the signal-to-noise ratio in most characterizations is substantially improved because of the large number of layers being measured. In this way, SQUID is in effect measuring the magnetism of gZCO monolayers with a mass comparable to the bulk, overcoming the above-mentioned challenge. M-H curves of gZCO ($x = 0.119$)/rGO are collected (Fig. 1f, see Methods). We observe robust hysteresis loops with a coercive field of 840 Oe at 2 K and 280 Oe at 300 K; this provides direct evidence of ferromagnetism in 2D gZCO. Considering the hysteresis loops are still prevalent up to 300 K, the Curie temperature of gZCO must be even higher. In the meantime, an in-plane magnetic anisotropy is unveiled because of a smaller saturation field required in the ab plane than along the c-axis (Supplementary Fig. S2). It is worth mentioning that the considerable coercive fields completely exclude super-paramagnetic nanoparticles in the gZCO system, such as Co and cobalt oxides, which yield a vanishing coercivity[33]. TEM also rules out clusters due to Co doping. Figure 1g reveals a cross-sectional view of gZCO/rGO via energy-dispersive X-ray spectroscopy (EDS) mapping. Co atoms are sparsely and randomly distributed, displaying no clustering. A high-resolution TEM (HRTEM) image of gZCO/rGO with the corresponding intensity profile is plotted in Fig. 1h, i. The layers of gZCO are clearly resolved with monolayer gZCO intercalated into the vdW gaps of rGO.

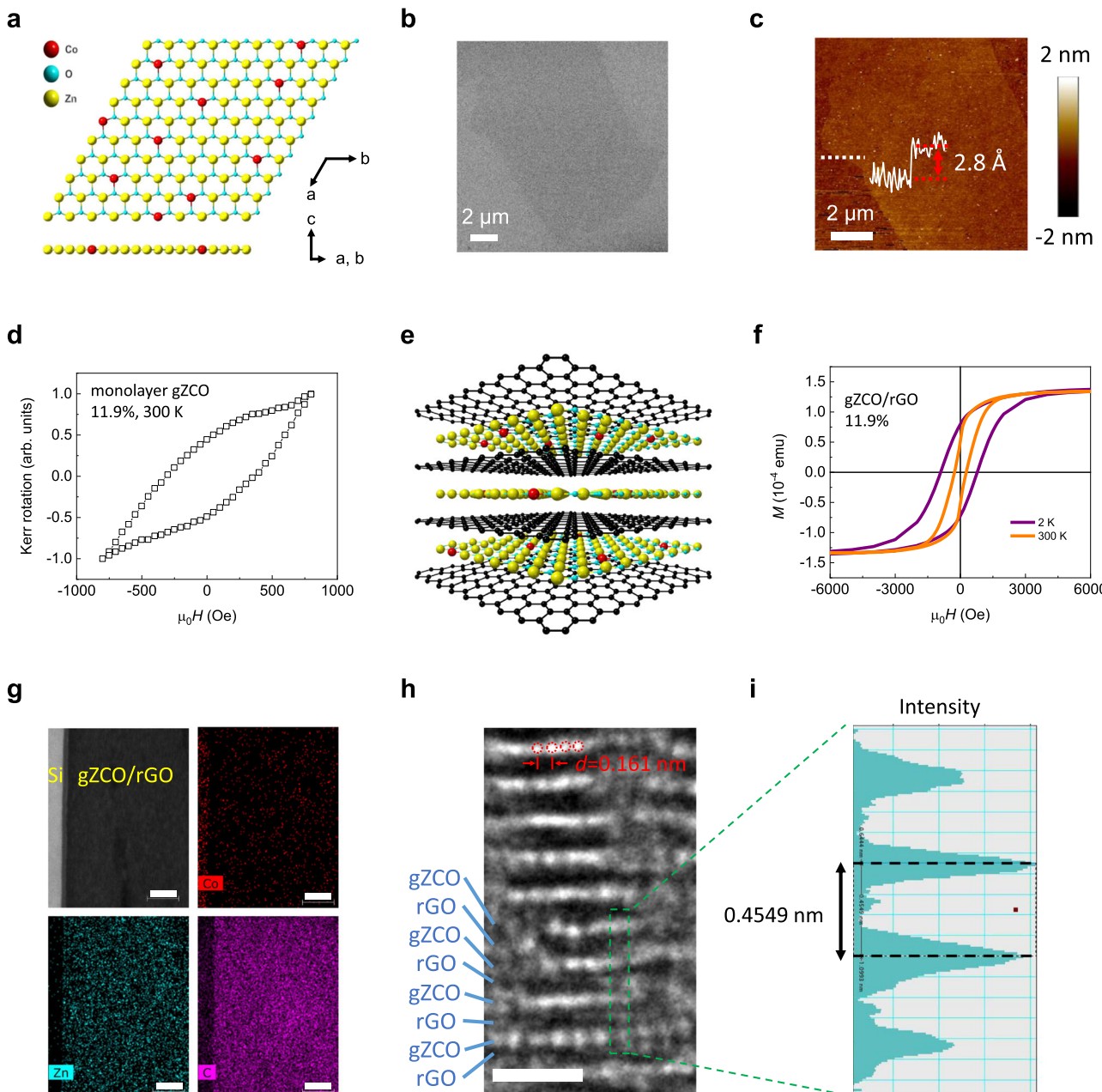

**Fig. 1 2D vdW gZCO. a** Schematic crystal structure of Co-doped monolayer gZnO displaying a characteristic honeycomb structure. Red, blue, and yellow spheres denote Co, O, and Zn atoms, respectively. **b**, **c** Atomically uniform and clean morphology of a monolayer gZCO sheet on a 100 nm SiO$_2$/Si substrate via the SEM (**b**) and AFM (**c**) imaging, distinctly eliminating the existence of clustered particles on the sample surface. The AFM cross-sectional plot along the white dashed line in (**c**) shows the single-layer thickness around 2.8 Å. Scale bar: 2 μm. **d** Longitudinal MOKE of a monolayer Zn$_{0.881}$Co$_{0.119}$O sheet at room temperature, displaying typical ferromagnetic hysteresis loops. The magnetic field is swept between ±800 Oe in the sample plane. The monolayer nature, together with ultra-clean sample surfaces, is clearly verified by AFM scanning, as shown in Supplementary Fig. S1a. **e** Schematic illustration of an alternately stacked rGO and gZCO heterostructure that occurs during the synthesis process. The narrow vdW gaps of rGO stabilize the 2D form of gZCO. Such heterostructure is widely used in the following characterizations to significantly enhance the mass load of gZCO, as well as the signal-to-noise ratio. Black sphere: C atom. **f** Hysteretic *M-H* loops acquired by SQUID showing solid ferromagnetic long-range order in 2D Zn$_{0.881}$Co$_{0.119}$O at 2 and 300 K. The external magnetic fields are swept parallel to the sample plane. **g** EDS mapping of the cross-sectional gZCO/rGO on the Si substrates, which confirms the diluted Co doping concentration, and eliminates the clustered Co or cobalt oxides. Scale bar: 10 nm. **h**, **i** Side view of gZCO/rGO via HRTEM (**h**) along with the corresponding intensity profile (**i**). The white dots are corresponding to the gZCO layers while the dark gap area corresponds to rGO. gZCO atomic layers are clearly shown with the lattice parameter of {11$\bar{2}$0} plane being 0.161 nm. The interlayer spacing is enlarged to 0.4549 nm owing to the existence of rGO layer between two adjacent gZCO layers. Scale bar: 1 nm.

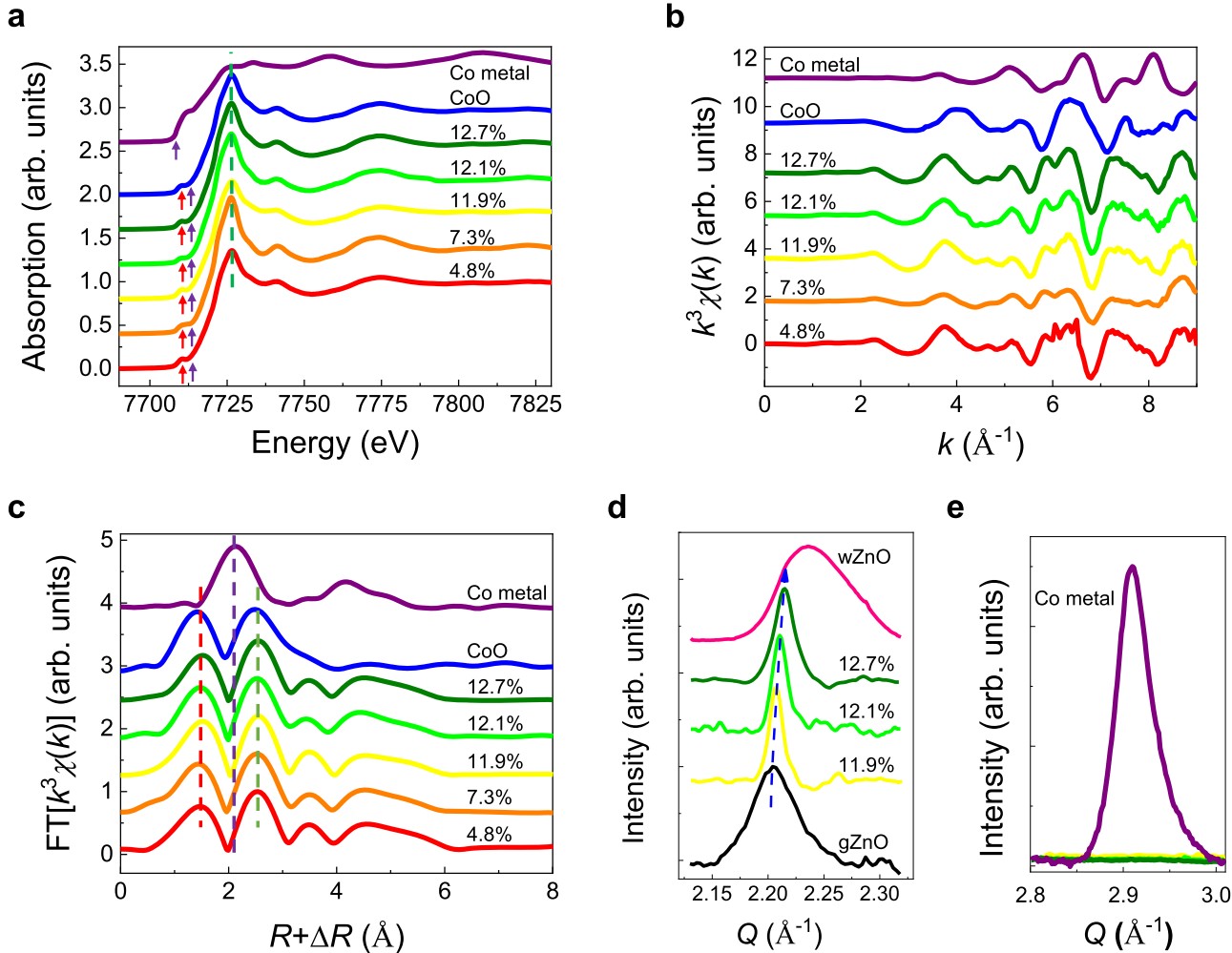

**Fig. 2 X-ray characterizations of 2D gZCO. a** XANES of dispersion of gZCO/rGO alternate structures at Co $K$-edge as a function of Co concentration, showing representative pre-edges (red arrows), rising edges (purple arrows), white lines (green dashed line), and continua of $Co^{2+}$ states. **b** $k^3$-weighted EXAFS at Co $K$-edge in gZCO/rGO. **c** Fourier transformed EXAFS spectra in (**b**) resolving the atomic coordination shells. The red and green dashed lines denote the nearest-neighboring Co-O and Co-Zn distances in 2D gZCO, while the purple one marks the nearest-neighboring Co-Co bonding length in metallic Co. The above characteristic $Co^{2+}$ features of 2D gZCO in (**a**–**c**) differ drastically from those in the reference Co metals. **d**, **e** GIWAXS plots of various 2D gZCO/rGO samples in the vicinity of the $\{10\bar{1}0\}$ lattice plane regions of ZnO (**d**) and metallic Co (**e**). The lattice constants of $\{10\bar{1}0\}$ plane are 2.846, 2.843, and 2.837 Å when $x = 0.119$, 0.121, and 0.127, respectively, which are closer to that of gZnO (2.85 Å)[23–25], while much smaller than that of wurtzite ZnO (wZnO, 2.81 Å). The above evidence suggests the highly crystalline 2D graphitic structures of pristine ZnO hosts, free of 3D wurtzite phase separation, metallic Co clusters, lattice deformation, and so on.

Since the magnetic interactions between doped Co atoms are critically dependent on the form of the dopants, it is imperative to verify the valence states and atomic coordination structures of Co. To this end, we carry out X-ray absorption spectroscopy (XAS) characterizations at the Co $K$ absorption edge. Both X-ray absorption near edge structure (XANES) and extended X-ray absorption fine structure (EXAFS) provide strong support for the incorporated $Co^{2+}$ ions in the gZnO lattice. First, distinct pre-edge features occur in gZCO, which would otherwise be forbidden in Co metal (red arrows in Fig. 2a). This is ascribed to hybridization between $d$ states of Co and $p$ states of O in gZCO, giving rise to an electric-dipole allowed $1s$ to $2p$ transition[19,34]. Second, the rising edge of gZCO is in good accordance with that of Co-O bonding, and a blueshift from Co metal to gZCO can be interpreted by +2 oxidation states (purple arrows in Fig. 2a)[19,34]. Third, the white line (green dashed line in Fig. 2a) and the subsequent oscillations of gZCO differ significantly from the behavior in Co metals. Finally, the striking differences between gZCO and Co metal are clearly resolved by the $k^3$-weighted $\chi(k)$

function (Fig. 2b) along with the scattering paths obtained after Fourier transformation. It is notable that the absence of Co-Co coordination shells at 2.117 Å unambiguously rules out metallic Co clusters in gZCO (purple dashed line in Fig. 2c)[19].

To clarify the crystal structures, we perform grazing-incidence wide-angle X-ray scattering (GIWAXS) tests to probe the in-plane lattice constant. Figure 2d illustrates the momentum transfer vector of the $\{10\bar{1}0\}$ plane in gZCO. After $Co^{2+}$ incorporation, the ultra-thin gZCO sheet retains a 2D vdW graphitic phase, with $\{10\bar{1}0\}$ lattice spacing consistent with that of gZnO[23–25]. A slight redshift of the in-plane lattice spacing with increasing Co doping manifests a smaller ionic radius of $Co^{2+}$ than $Zn^{2+}$[35]. In addition, GIWAXS analyses of gZCO prove the absence of metallic Co (Fig. 2e)[15]. With the above solid evidence combined, we confirm that Co doping in 2D gZnO is an ideal substitution of Zn as opposed to metallic phase segregation.

MOKE measurement results on 2D gZCO and various thicknesses of Co thin films also show clear distinctions. The ferromagnetic hysteresis loop obtained by MOKE on a bilayer gZCO

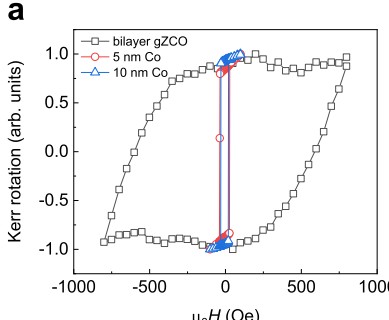
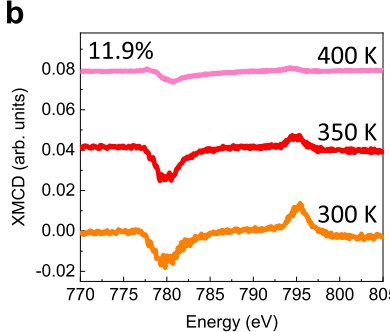
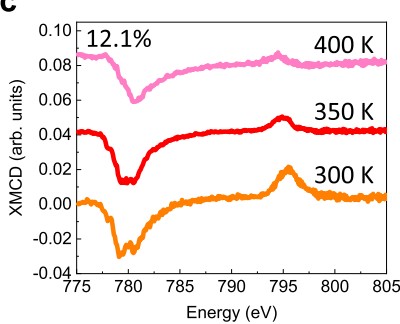

**Fig. 3 Room-temperature ferromagnetism in 2D gZCO. a** Longitudinal MOKE of a bilayer $Zn_{0.879}Co_{0.121}O$ sheet at 300 K, as well as 5- and 10-nm Co thin films for reference. The bilayer feature and atomically uniform sample surface are identified by AFM imaging (Supplementary Fig. S1b). **b, c** Temperature-dependent XMCD spectra of alternately stacked 2D gZCO/rGO heterostructures at Co $L_{2,3}$ edges for 11.9% (**b**) and 12.1% (**c**) doping concentration, respectively. XMCD is the difference between the two XAS spectra with magnetization parallel and antiparallel to the X-ray incident direction, and hence a non-zero XMCD is arising from the ferromagnetic 2D gZCO. Also, the Co $L_3$ edge position (780.5 eV at 400 K) of XMCD indicates the strongly entangled $Co^{2+}$ as the ferromagnetic origin.

($x = 0.121$) is shown in Fig. 3a, with a 600 Oe coercive field. As a control, the hysteresis loops of 5- and 10-nm Co thin films measured with the same setup are also plotted in Fig. 3a, which exhibit coercive fields that are nearly 20 times smaller. This stark contrast clearly eliminates ferromagnetism induced by clustered Co in gZCO. MOKE measurements are also done on the $SiO_2$/Si substrate outside the sample area, which shows trivial diamagnetic features with zero coercivity.

We also use synchrotron X-ray magnetic circular dichroism (XMCD) spectra at Co L-edges to explore the ferromagnetic origin (see Methods). Utilizing a total electron yield mode, we are capable of recording the XMCD in an ultra-thin region, merely 10 nm deep from the top sample surface[36]. Figure 3b, c shows the XMCD of 2D gZCO/rGO alternate structures at 300, 350, and 400 K. Non-vanishing XMCD features indicate ferromagnetic long-range order at 400 K and above. The prominent dips at 400 K are centered at 780.5 eV for both 11.9% and 12.1% Co doping (Fig. 3b, c), consistent with the Co-$L_3$ edge of the divalent states[20,21]. Therefore, the ferromagnetic origin of gZCO is explicitly attributed to the strongly coupled $Co^{2+}$ ions in the gZnO matrix instead of Co clusters, which agrees well with the above AFM, TEM, XAS, GIWAXS, and MOKE analyses. Meanwhile, on the basis of the spin and orbital sum rules[37], we calculate quantitatively the saturated magnetic moment $M_s$ per Co atom at 300 K from the XMCD spectra: 0.68 $\mu_B$/Co for 2D $Zn_{0.881}Co_{0.119}O$ and 0.93 $\mu_B$/Co for 2D $Zn_{0.879}Co_{0.121}O$, respectively. BMP was proposed to ferromagnetically align these spin moments. However, some factors may lead to a deviation from the ideal magnetization value, such as non-ideal Co doping and oxygen vacancy concentration, isolated Co ions, misalignment between Co minority band and impurity band, superexchange interaction, and so on. The magnetization values obtained above are comparable to the previously reported ones (about 0.07–2 $\mu_B$/Co) in the 3D ZCO system[18], which suggests a general magnetization range of the DMO systems as well as a similar physical mechanism of the magnetic couplings between 2D and 3D DMOs. Such considerable $M_s$ does not originate from super-paramagnetic or weakly ferromagnetic cobalt oxide particles, which would otherwise be ~0.01 $\mu_B$/Co[15].

To further exclude the influence of Co or cobalt oxides clusters, we conduct SQUID tests of a few control samples including pure gZnO/rGO (Supplementary Fig. S4) and samples prepared with pure Co precursors (Supplementary Fig. S5). The latter ones are intended to produce large numbers of Co and cobalt oxides clusters if there would be any of them in the gZCO/rGO composite. Both types of the above samples display typical

paramagnetic properties, distinctive from 2D gZCO. In the meantime, a linear volume dependence of the saturated magnetic moments is obtained in 2D $Zn_{0.881}Co_{0.119}O$/rGO, as shown in Supplementary Fig. S6. In light of the above evidence, it should be emphasized that we entirely exclude ferromagnetism in 2D gZCO contributed by the gZnO host, Co or cobalt oxides nanoparticles, and exterior contaminations. Namely, it is the substitutional $Co^{2+}$ ions in the gZnO matrix that are spontaneously ordered and give rise to robust ferromagnetic ground states.

To probe the tunable magnetism, we produced a series of samples with various doping levels. Figure 4a summarizes the tunable coercive fields when the Co concentration varies. Pristine gZnO ($x = 0$) has zero coercivity in $M$-$H$ curves and its $M$-$T$ behavior fits well with the Curie–Weiss law (Supplementary Fig. S4), which is typical for paramagnets. At higher doping levels such as $x = 0.048$ (Fig. 4b) and 0.073 (Fig. 4c), gZCO turns into a weak ferromagnet with the occurrence of $M$-$H$ hysteresis loops. As we continue to increase the doping levels, maximum coercive fields are observed in $Zn_{0.881}Co_{0.119}O$ (Fig. 1f) and $Zn_{0.879}Co_{0.121}O$ (Fig. 4d), which, in turn, indicates optimal ferromagnetism in such series of samples. If we further increase the doping level, however, compared with $Zn_{0.879}Co_{0.121}O$, the ferromagnetic behavior starts to decline as the coercive field drops (Fig. 4e) and magnetic moments are anomalously compressed at low temperatures (Supplementary Fig. S7). We attribute such suppressed ferromagnetic behavior to a less-ordered ground state.

## Discussion

In principle, it has been proposed that the BMP-induced impurity-band-exchange interaction is responsible for ferromagnetism in most wide-bandgap DMOs[18]. When bound polarons stemming from the carrier donors percolate, they will mediate a long-range ferromagnetic exchange coupling among the doped magnetic atoms. Another important factor is the concentration of Co. We assume that one polaron site can ferromagnetically interact with spins within a range of about ten atom sites considering a hydrogenic orbital of localized carriers[18]. Therefore, we expect that a long-range ferromagnetic order will occur at the Co doping level around (percolation threshold)/10. In 2D case, the percolation threshold for the triangular lattice is 50%[38], which indicates a requirement of 5% Co to achieve ferromagnetism, and the ferromagnetism will be further enhanced with higher Co occupancy. Furthermore, when Co ions percolate (or just come close enough), the oxygen anions will mediate a superexchange interaction that prefers antiferromagnetic ordering, which competes with the BMP-induced ferromagnetic

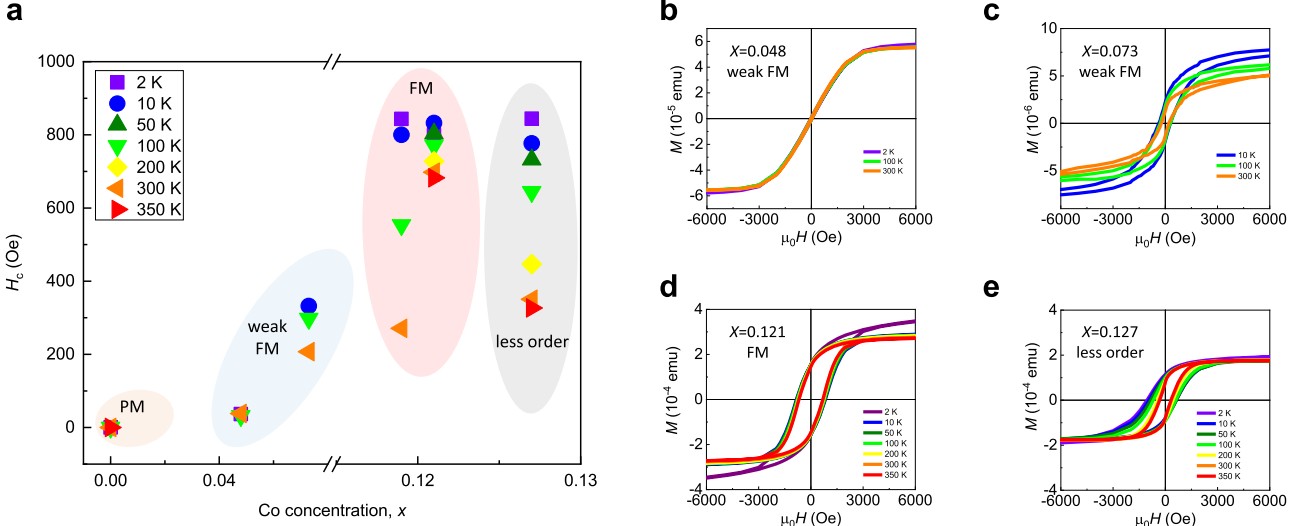

**Fig. 4 Tunable magnetism in gZCO. a** Summary of temperature- and Co-concentration-dependent coercive fields in gZCO obtained by SQUID. With increased Co concentrations, gZCO can be tuned from paramagnetic (PM) to ferromagnetic (FM), and to less-ordered states. **b–e** $M$-$H$ curves of $Zn_{0.952}Co_{0.048}O/rGO$ (**b**), $Zn_{0.927}Co_{0.073}O/rGO$ (**c**), $Zn_{0.879}Co_{0.121}O/rGO$ (**d**), and $Zn_{0.873}Co_{0.127}O/rGO$ (**e**), on the basis of SQUID measurements. From (**b**) to (**d**), the ferromagnetism becomes stronger. Optimal ferromagnetism has been realized when $x = 0.121$, with a pronounced coercive field 800 Oe at 2 K and 690 Oe at 350 K. When $x = 0.127$, a less-ordered magnetic state occurs, as evidenced by the shrunk coercive fields (**e**) and the suppressed magnetic moments at cryogenic temperatures (Supplementary Fig. S7b). Meanwhile, a higher magnetic field can be used to stabilize long-range order (Supplementary Fig. S7c). The less-ordered state can be explained by an interesting competition between ferromagnetic impurity-band-exchange interaction and antiferromagnetic superexchange interaction.

exchange interaction. Therefore, when raising the doping level of magnetic atoms, the ground state of gZCO could undergo a transition from ferromagnetism to an antiferromagnetism. When the Co doping exceeds $x = 0.121$, the magnetization starts to decline, suggesting that there could be competing mechanisms in the system. The onset of short-range antiferromagnetic super-exchange interactions competes with the ferromagnetic spin configuration and gives rise to less-ordered magnetic behaviors. This promises a strong tunability of magnetization by changing the doping level[18,22]. More specifically, the cation percolation threshold is 50% in a 2D triangular lattice[38], which is much higher than 18% in 3D wurtzite ZnO[18]. Therefore, a higher doping concentration of Co is required in 2D ZCO to frustrate the long-range ferromagnetic order, which yields the occurrence of optimal ferromagnetism in a higher doping region than 3D (less than 10%)[17,18]. On the other hand, by applying the 3D BMP-induced ferromagnetism[18] to 2D through a straightforward dimensionality reduction, we also roughly estimate the Curie temperature $T_C$ of 2D ZCO, which is comparable to or even higher than its 3D counterparts (see Methods). Our experimental observation of room-temperature ferromagnetism and magnetic phase transition is consistent with the BMP theory, which may be helpful in verifying the validity of this model. More in-depth theoretical studies in the future will be helpful to better understand the change of Curie temperature and BMP hypothesis at the 2D limit.

In summary, we have realized Co substitution in monolayer and bilayer gZnO where long-range ferromagnetic order is demonstrated at room temperature and with environmental stability. Our results also highlight the potential to tune the magnetism in 2D materials by changing the doping level of magnetic atoms. Compared with other 2D ferromagnetic candidates, gZCO marks a crucial step forward in the realization of spintronics, magneto-optics, as well as new quantum and topological phases[40–41] at ambient conditions. Meanwhile, the atomically thin geometry of gZCO opens an unprecedented avenue to explore the underlying spin coupling mechanisms in DMO

systems, by making all magnetic atoms accessible with advanced probes such as spin-polarized scanning tunneling microscopy.

## Methods

**Sample synthesis**. Atomically thin gZCO crystals are synthesized via a solution-based and template-assisted method. Commercially available GO nanoflakes (Graphene Laboratories Inc.) are dispersed in aqueous solution, which are then drop-casted onto 100 nm $SiO_2/Si$ substrates. After drying at 40 °C, the GO layers are immersed into a mixture precursor solution consisting of Zinc acetate dihydrate (99.999% pure, Sigma-Aldrich) and Cobalt acetate dihydrate (99.999% pure, Sigma-Aldrich) for at least 8 h, during which the precursors are spontaneously intercalated into the GO vdW gaps. The concentration ratio between Zn and Co in the precursor determines the eventual stoichiometry in gZCO. Once the intercalation is finished, the templates are rinsed by deionized water for no less than 5 min to entirely remove the precursors on the sample surface. Precursors encapsulated by GO nanoflakes are annealed at 625 °C in vacuum to form alternating gZCO/rGO heterostructures. It is worth noting that these unique alternate structures are widely used in this paper for magnetic, electronic and structural characterizations, where gZCO keeps its 2D form and the mass load is significantly enhanced. In the end, the heterostructures are baked in air at 400 °C for 1 h to remove rGO, subsequently forming gZCO atomic layers. A thinner GO template tends to yield thinner gZCO layers. This annealing process also highlights the surprisingly high-temperature robustness of the gZCO system in the air.

**Magnetic characterizations**. Room-temperature longitudinal MOKE of monolayer and bilayer gZCO sheets are carried out in an Evico magnetics Kerr-Microscope & Magnetometer system with a blue LED light source. An in-plane magnetic field is generated with a rotatable electromagnet. The integration of the gray level on the sample region provides the longitudinal Kerr rotation.

The magnetic moments of alternatingly stacked 2D gZCO/rGO are measured in a Quantum Design MPMS-XL SQUID magnetometer using vibrating sample magnetometer mode as a function of magnetic field and temperature. gZCO encapsulated by rGO is prepared into the form of thin films, with lateral size of a few $mm^2$ and thickness of μm level (identified by a step profiler). The quantity of 2D gZCO sheets is substantially increased to reach the sensitivity of SQUID. Kapton tapes are used to peel off such gZCO/rGO thin films, which are then mounted onto a plastic straw holder for the SQUID tests. $M$-$H$ curves are acquired with an external magnetic field sweeping between ±1 T, while the $M$-$T$ curves are obtained after a field cooling process to fully polarize the spins in the gZCO plane. For the $M$-$T$ measurements, a relatively small field of 100 or 1000 Oe is used to approach the intrinsic spin behaviors. Magnetic fields are all applied in the sample plane during the SQUID measurements.

XMCD characterizations are performed at beamline 6.3.1 at the Advanced Light Source. The X-ray exhibits a 30° incidence angle between beam and gZCO/rGO

sample surface, which enables the detection of in-plane ferromagnetic signals. XAS spectra are recorded at Co $2p$ levels (from 765 to 815 eV) with the incident X-rays circularly polarized. An external magnetic field of 0.6 T is applied parallel and antiparallel to the X-ray helicity; thus, the difference between the two XAS spectra is defined as XMCD. A non-zero XMCD indicates a ferromagnetic ground state. Then the gZCO/rGO samples are heated from 300 to 350 K and then to 400 K, to explore the magnetic behaviors beyond room temperature. Note that the total electron yield mode merely probes a 10-nm-deep region from the sample surface, which displays remarkable sensitivity for the detection of 2D ferromagnets. From the acquired XMCD spectra, a detailed calculation of spin and orbital moments in 2D gZCO is on the basis of the sum rules[37]. The equations are as follows:

$$m_{\text{orb}} = -\frac{4\int_{L_3+L_2}(\mu_+ - \mu_-)d\omega}{3\int_{L_3+L_2}(\mu_+ + \mu_-)d\omega}(10 - n_{3d}), \tag{1}$$

$$m_{\text{spin}} = -\frac{6\int_{L_3}(\mu_+ - \mu_-)d\omega - 4\int_{L_3+L_2}(\mu_+ - \mu_-)d\omega}{\int_{L_3+L_2}(\mu_+ + \mu_-)d\omega} \times (10 - n_{3d})\left(1 + \frac{7\langle T_z \rangle}{2\langle S_z \rangle}\right)^{-1}. \tag{2}$$

Here $m_{\text{orb}}$ and $m_{\text{spin}}$ (in the unit of $\mu_B$/Co) denote the contributions from orbital and spin to magnetic moments. $L_3$ and $L_2$ are the integration region at the corresponding Co $L$ edges. The XAS spectra are represented by $\mu_+$ and $\mu_-$ when magnetization is parallel and antiparallel to the X-ray helicity, respectively. $\omega$ is the photon frequency (energy). Note that $n_{3d}$ is the occupied number of $3d$ electrons of Co in gZCO. $\langle T_z \rangle$ is the expectation value of the magnetic dipole operator and $\langle S_z \rangle$ equals to half of $m_{\text{spin}}$ in Hartree atomic units, the ratio between which is reported to be negligible in the Co systems[37]. We calculate the saturated magnetic moments by adding up $m_{\text{orb}}$ and $m_{\text{spin}}$.

**Material characterizations**. The morphology of ultra-thin gZCO sheets is studied using SEM (Zeiss Gemini Ultra-55 Analytical SEM) and AFM (Veeco D3100) tapping modes. The cross-sectional gZCO/rGO samples for (HR)TEM are prepared by focused ion beam by following the standard procedure and showered by 5 kV Ga ion beam to remove the damage introduced by 30 kV Ga ion beams. The EDS mapping was performed on a FEI Talos operating at 200 kV equipped with a SuperX EDS. The HRTEM was taken by ACAT at 200 kV. X-ray photoelectron spectroscopy (XPS) measurements (Supplementary Fig. S3) are conducted to explore the valence states of Co at $2p$ core levels (775–815 eV) from a PHI 5600 XPS using a 2 mm Al monochromatic source at 15 kV and 350 W with a spot size of $400 \times 400$ μm². Also, we obtain the Co doping concentrations by integrating and comparing the XPS spectra at Co and Zn $L$ edges. XAS characterizations are performed at the Stanford Synchrotron Radiation Lightsource (SSRL). Spectra are collected at Co $K$-edge (from 7480 to 8065 eV) in a fluorescence mode with a 45° incident angle between the X-ray beam and the sample surface (beamline 4-1). GIWAXS data are recorded at beamline 11-3 at SSRL. Scattering patterns are measured with a grazing angle of about 0.18° for the detection of in-plane structural information. The sample chamber is filled with Helium during the measurement. We are using gZCO/rGO alternate structures for the above XPS, XAS, and GIWAXS characterizations.

**Theoretical calculation**. We apply the 3D BMP model[18] to 2D through a straightforward dimensionality reduction (e.g., the volume $4\pi r^3/3$ becomes the area $\pi r^2$). We obtain the Curie temperature as

$$T_C = \frac{\pi n_O r_c^2 J_{sd}}{2k_B}\sqrt{\frac{S(S+1)}{3}x\delta}, \tag{3}$$

where $n_O = 1.09 \times 10^{19}$ m⁻² is the areal density of oxygen atoms (volume density $n_O = 6 \times 10^{28}$ m⁻² for the 3D case), $r_c$ is the cation radius (0.06 nm), $J_{sd}$ is the $s$-$d$ exchange interaction between impurity-band carriers and localized $Co^{2+}$ ions, $\delta = n_{\text{vac}}/n_O$ is the percentage of oxygen vacancies per unit cell, $S$ is the spin of $Co^{2+}$ ions, and $k_B$ is the Boltzmann constant. If we only take into consideration the geometry change, we can roughly compare the $T_C$ between 2D and 3D cases: $\frac{T_{c,2D}}{T_{c,3D}} = \frac{3}{4}\left(\frac{n_{0,2D}}{n_{0,3D}}\right)\frac{1}{r_c} > 1$. This indicates that the Curie temperature of 2D ZCO can be at least comparable to its 3D counterparts. Other parameters in the model may also be affected by the dimensionality. For instance, from 3D to 2D, the quantum confinement and structural phase transition can induce a variation of hybridization between donor impurity band and Co $3d$ orbitals, which will affect the strength of the $s$-$d$ exchange parameter $J_{sd}$, and thus the Curie temperature. More in-depth theoretical calculations in the future will provide a better understanding of the Curie temperature and BMP hypothesis at the 2D limit.

## Data availability
The data that support the findings of this study are available from the corresponding authors upon reasonable request.

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

## Acknowledgements

We warmly thank Dr. Yimeng Gu, Dr. Hongrui Zhang, Dr. Shuren Lin, and Dr. Kyle B. Tom for the insightful discussions. This work is funded by the Intel Corporation under an award titled Valleytronics center and the Bakar Fellows Program at University of California, Berkeley. Y. S., S. W., X. C., R. J. B., F.Y., and Q.W. acknowledge the support of the US Department of Energy, Office of Science, Office of Basic Energy Sciences, Materials Sciences and Engineering Division under Contract No. DE-AC02-05-CH11231 within the Quantum Materials Program (KC2202) and Organic-Inorganic Nano-composites Program (KC3104). J. H. and H. Y. acknowledge the support by the National Natural Science Foundation of China (51861145201, 91750101, 21733001). This research used resources of the Advanced Light Source, which is a DOE Office of Science User Facility under contract no. DE-AC02-05CH11231. Use of the Stanford Synchrotron Radiation Lightsource, SLAC National Accelerator Laboratory, is supported by the US Department of Energy, Office of Science, Office of Basic Energy Sciences under Contract No. DE-AC02-76SF00515. Work at the Molecular Foundry was supported by the Office of Science, Office of Basic Energy Sciences, of the US Department of Energy under contract no. DE-AC02-05CH11231. Use of the Center for Nanoscale Materials, an Office of Science user facility, was supported by the US Department of Energy, Office of Science, Office of Basic Energy Sciences, under Contract No. DE-AC02-06CH11357. We acknowledge the Biomolecular Nanotechnology Center for access and assistance with measurement systems.

## Author contributions

R.Chen. and J.Y. conceived the project and designed the experiments. R.Chen., F.L., F.Y., Z.F., and Q.W. synthesized the samples. R.Chen., Y.S., S.W., and X.C. performed SQUID measurements. R.Chen., Y.D., J.H., and H.Y. worked on the MOKE microscopy. R.Chen., J.C., A.N., and P.S. obtained XMCD spectra. R.Chen. and F.L. carried out AFM, XAS, and GIWAXS characterizations. Y.Z.L., R.Chen., S.L., Y.L., and D.J. contributed to the electron microscopy. R.Cheng., R.Chen., J.Y., and R.J.B. conducted the theoretical analyses. R.Chen. and J.Y. wrote the manuscript. All authors discussed the results and commented on the manuscript.

## Competing interests

The authors declare no competing interests.
