## [Peer Review File · Nature Communications]

Reviewers' Comments:

Reviewer #1:

Remarks to the Author:

The authors reported the room temperature ferromagnetism in Co-doped two-dimensional semiconductor ZnO by magneto-optic Kerr effect microscopy(MOKE) and superconducting quantum interference device(SQUID). They conducted a series of experiments to exclude the possible contribution of metallic Co or cobalt oxide clusters. They also use X-ray magnetic circular dichroism(XMCD) to confirm divalent state of substitutional Co atoms. This experimental result of room temperature ferromagnetism in Co-doped graphene-like ZnO follows recent trends. However, I can not agree to publish this article since it lacks strong evidence to me both in experimental and theoretical points of view. I suggest the authors need to give both more solid evidence from experimental and theoretical and qualitative analysis from 3D to 2D transition in the article.

1. The authors explained the ferromagnetism by citing the reference of bound magnetic polaron induced impurity-band exchange mechanism. The model was used to describe the ferromagnetism in three dimensional dilute magnetic semiconductors in which both the disorder and interactions must be taken into account. The disorder-induced localization effect will be different in different dimensions. What is the influence of dimensionality on the magnetic properties of two-dimensional doped semiconductors such as Curie temperature and optimized doping density. At least a qualitative dimensionality analysis of magnetism should provide.

2. The author mentioned in the first paragraph: "Non-Heisenberg interactions such as magnetic anisotropy lift the Mermin-Wagner constraint and enable long-range magnetic order". What is the exact meaning for Non-Heisenberg interactions? Long-rang interaction or short-range interaction? Any experimental evidence for the Non-Heisenberg interactions? Usually, an on-site uniaxial magnetic anisotropy is necessary for long-range magnetic order in two dimensions. The presence of easy-plane anisotropy only permits quasi-long range order, as described in 2D XY model.

3. What is the easy-axis of the Co-doped 2D graphene-like ZnO? The authors only mentioned the applied magnetic field is in the film plane. However what is the anisotropy of Co-doped 2D ZnO?

4. Some previous articles mentioned that ZnCoO is not a ferromagnetic material, Co(II) ions are isolated from each other, and consequently, the films are paramagnetic, such as Journal of Applied Physics 104, 033901 (2008). What is the fundamental physics provide the long-range ordering in bilayer and monolayer? Also the Curie temperature is room temperature and this suggests the ordering strength is large comparing with the dipolar interaction from Co ?

5. Also it is well-known that CoO/Co core can induce exchange bias and thus the strong coercivity, such as JMMM Volume 303, Issue 2, August 2006, Pages e160-e164. Can authors also compare the CoO/Co core-shell hysteresis in Figure 3a to make sure no contribution from the oxidation of Co nanoparticles?

Reviewer #2:

Remarks to the Author:

The paper presents a wealth of experimental results on an innovative and challenging material. The proposed material is of particular interest due to its new concept that could open new avenues in the search for room temperature ferromagnets suitable for device applications.

The experimental results presented span a wide range of techniques used (XPS, SEM, AFM, TEM, EXAFS, XANES, GIWAXS) to characterize the structural properties of the system, leaving no room, to my opinion, for doubts on the perfectly clean layered structure of the samples.

In addition, the magnetic characterization is also very exhaustive and complete on the basis of MOKE, SQUID and XMCD results, able to rule out contributions from Co clusters and other spurious structures.

Finally, the authors were able to convincingly show the high tunability of the system acting on

doping as well as on volume.

Just on this last point, I have a remark for the authors: I think it would be interesting if they could comment a little more on the different samples (#1-#5) shown in Fig. S5 and demonstrating a linear behavior of the saturation magnetization vs volume of the sample: information regarding how the volume is changed (number of layers of gZCO, sample size, planar size..) would be very helpful.

To conclude, I think that the paper is overall very interesting, well written, and discusses original and solid results that could be of wide interest for the community.

Alessandra Continenza

Reviewer #3:

Remarks to the Author:

Two-dimensional (2D) van der Waals (vdW) magnetic materials are of current great interest for their promising spintronic applications. However, the ferromagnetic ordering temperature is far below room temperature (RT) for those insulating or semiconducting systems. This question is now answered by the present work, which reports tunable RT ferromagnetism in Co-doped 2D vdW ZnO. This work uses transmission electron microscopy and atomic force microscopy results to exclude the existence of metallic Co or cobalt oxides clusters in the Co-doped 2D graphene-like ZnO (gZCO). The authors have observed a spontaneous magnetization in gZCO at room temperature and above, through the magneto-optic Kerr effect, superconducting quantum interference device and X-ray magnetic circular dichroism measurements. The x-ray characterizations suggest that the substitutional Co atoms are in the +2 charge state. Then the authors assume that the substitutional Co²⁺ ions form the bound magnetic polarons and are responsible for the RT ferromagnetism via the impurity-band-exchange interactions.

This is an interesting work and presents important experimental results. Before it can be further considered for publication in Nature Communications, the following points may be addressed by the authors.

1. How can the interstitial Co atoms be excluded? How about the oxygen stoichiometry? Are there oxygen vacancies to introduce the carriers? Otherwise, how would the doped Co²⁺ ions be strongly coupled to form RT ferromagnetism?
2. The doped Co²⁺ ions are most likely in the high-spin state due to the low coordination and to the weak crystal field. They have each the spin moment of 3 μ B. As the present gZCO is claimed to be a RT ferromagnet, why does the magnetization value of 0.68-0.93 μ B/Co become much smaller?
3. Besides a set of experimental results showing the RT ferromagnetism, a theoretical input may be expected. The authors have briefly mentioned the picture of bound magnetic polarons and impurity-band-exchange, but the present discussion is still far from being clear. How the doped magnetic impurities form a RT ferromagnetic order in a wide gap semiconductor seems to be a standing issue. The readers may also like to see more theoretical input in the present work.

Point-by-point response to Reviewers' comments:

Reviewer #1 (Remarks to the Author):

The authors reported the room temperature ferromagnetism in Co-doped two-dimensional semiconductor ZnO by magneto-optic Kerr effect microscopy(MOKE) and superconducting quantum interference device(SQUID). They conducted a series of experiments to exclude the possible contribution of metallic Co or cobalt oxide clusters. They also use X-ray magnetic circular dichroism(XMCD) to confirm divalent state of substitutional Co atoms. This experimental result of room temperature ferromagnetism in Co-doped graphene-like ZnO follows recent trends. However, I can not agree to publish this article since it lacks strong evidence to me both in experimental and theoretical points of view. I suggest the authors need to give both more solid evidence from experimental and theoretical and qualitative analysis from 3D to 2D transition in the article.

Our response:

We thank the reviewer very much for reviewing our manuscript and considering our results interesting. Please find our detailed response to your comments as follows.

1. The authors explained the ferromagnetism by citing the reference of bound magnetic polaron induced impurity-band exchange mechanism. The model was used to describe the ferromagnetism in three dimensional dilute magnetic semiconductors in which both the disorder and interactions must be taken into account. The disorder-induced localization effect will be different in different dimensions. What is the influence of dimensionality on the magnetic properties of two-dimensional doped semiconductors such as Curie temperature and optimized doping density. At least a qualitative dimensionality analysis of magnetism should provide.

Our response:

We thank the reviewer for bring up this important point. The dimensionality effect on the disorder and hence the magnetic interactions is a fundamental theoretical question and we hope our experimental findings reported in this manuscript will add to the understanding and eventually help establish a comprehensive theory in the future. In the revised manuscript, we have included our preliminary discussions on this topic to provoke more in-depth theoretical analysis by colleagues in this field.

In 2D Co-doped ZnO (ZCO), a higher optimized doping density has been observed, in contrast with its 3D counterparts. Based on the bound magnetic polaron (BMP) model (Nature materials 4.2 (2005): 173-179; Physical Review Letters 88.24 (2002): 247202), the magnetic ground state is

determined by ferromagnetic impurity band exchange interaction (dominant at low doping level) and antiferromagnetic superexchange interaction (dominant at high doping level). The ferromagnetic response of the system is frustrated by the increasing antiferromagnetic interactions at higher doping levels. The doping level at which the transition happens is affected by the dimensionality. Comparing the 3D wurtzite phase to 2D graphitic phase of ZnO, the cation percolation threshold is 50% in a 2D triangular lattice (Introduction to percolation theory. CRC press, 2018), which is much higher than 18% in 3D wurtzite ZnO (Nature materials 4.2 (2005): 173-179). Therefore, a higher doping concentration of Co is required in 2D ZCO to frustrate long-range ferromagnetic order, which yields the occurrence of optimal ferromagnetism in a higher doping region than 3D. This interesting trend was explicitly verified by the SQUID measurements: the optimal doping density is $\sim 12.1\%$ in 2D ZCO (Fig. 4a in the manuscript), which turns out to be higher than that in 3D ZnO (less than 10%, Nature materials 4.2 (2005): 173-179; Physical review letters 93.17 (2004): 177206).

Another step we may take is to apply the 3D BMP model (Nature materials 4.2 (2005): 173-179, 202) to 2D through a straightforward dimensionality reduction (*e.g.* the volume $4\pi r^3/3$ becomes the area πr^2). We obtain the Curie temperature as

$$T_c = \frac{\pi n_o r_c^2 J_{sd}}{2k_B} \sqrt{\frac{S(S+1)}{3}} x \delta,$$

where $n_o = 1.09 \times 10^{19} m^{-2}$ is the areal density of oxygen atoms (volume density $n_o = 6 \times 10^{28} m^{-3}$ for the 3D case), r_c is the cation radius (0.06 nm), J_{sd} is the s-d exchange interaction between impurity-band carriers and localized Co^{2+} ions, $\delta = n_{vac}/n_o$ is the percentage of oxygen vacancies per unit cell, S is the spin of Co^{2+} ions, and k_B is the Boltzmann constant. If we only take into consideration the geometry change, we can roughly compare the T_c between 2D and 3D cases: $\frac{T_{c,2D}}{T_{c,3D}} = \frac{3}{4} \left(\frac{n_{o,2D}}{n_{o,3D}} \right) \frac{1}{r_c} > 1$. This indicates that the Curie temperature of 2D ZCO can be comparable to or even higher than its 3D counterparts. Other parameters in the model may also be affected by the dimensionality. More in-depth theoretical studies in the future will be helpful to better understand the Curie temperature and BMP hypothesis at the 2D limit.

The new analyses of dimensionality-dependent optimal doping level and Curie temperature have been added in the revised manuscript (Paragraph 1 on page 7 and paragraph 2 on page 18).

2. The author mentioned in the first paragraph: “Non-Heisenberg interactions such as magnetic anisotropy lift the Mermin-Wagner constraint and enable long-range magnetic order”. What is the exact meaning for Non-Heisenberg interactions? Long-rang interaction or short-range interaction? Any experimental evidence for the Non-Heisenberg interactions? Usually, an on-site uniaxial magnetic anisotropy is necessary for long-range magnetic order in two dimensions. The presence of easy-plane anisotropy only permits quasi-long range order, as described in 2D XY model.

Our response:

We appreciate the reviewer for the detailed comments.

By ‘non-Heisenberg interactions’, we were referring to magnetic anisotropies. Because of thermal fluctuations, 2D long-range magnetic ordering is prohibited by the continuous rotational symmetry and short-range interactions. Magnetic anisotropy, on the other hand, can overcome this limitation. This sentence generally refers to previous experimental studies on 2D ferromagnets such as Cr₂Ge₂Te₆, CrI₃, etc. (Nature 546.7657 (2017): 265-269; Nature 546.7657 (2017): 270-273) as a background, which are not directly related to what we are reporting in this manuscript. **To avoid confusion, we have rephrased this statement into ‘The introduction of magnetic anisotropy and other anisotropic interactions can lift the Mermin-Wagner constraint on the long-range magnetic order.’ (Paragraph 1, page 3)**

3. What is the easy-axis of the Co-doped 2D graphene-like ZnO? The authors only mentioned the applied magnetic field is in the film plane. However what is the anisotropy of Co-doped 2D ZnO?

Our response:

The easy axis of 2D ZCO is in the ab plane. In order to verify the magnetic anisotropy. Just like previous studies of 2D magnets (Nature 546.7657 (2017): 265-269; Nature 546.7657 (2017): 270-27), we obtained the *M-H* curves of 2D ZCO/rGO alternate layers at 300 K using SQUID. An external magnetic field was applied both in and perpendicular to the film plane successively (Fig. R1). A much smaller saturation field is required when the field is along the ab plane (± 1 T) compared to that of c-axis (± 4.5 T). Therefore, 2D ZCO displays a magnetic anisotropy with in-plane easy axis. Future efforts will be helpful to explore the anisotropy in the sample plane by using spin-polarized scanning tunneling microscopy (SP-STM).

The above analyses have been incorporated into Fig. S2 in the supplementary information.

Fig. R1 | M - H curves of 2D ZCO/rGO alternate layers at 300 K obtained by means of SQUID.

4. Some previous articles mentioned that ZnCoO is not a ferromagnetic material, Co(II) ions are isolated from each other, and consequently, the films are paramagnetic, such as Journal of Applied Physics 104, 033901 (2008). What is the fundamental physics provide the long-range ordering in bilayer and monolayer? Also the Curie temperature is room temperature and this suggests the ordering strength is large comparing with the dipolar interaction from Co ?

Our response:

We truly understand the concerns from the reviewer that some ZnCoO samples have been reported to be paramagnetic. However, in our work on 2D ZCO, room temperature ferromagnetism was unambiguously confirmed by independent measurements using MOKE, SQUID and XMCD. Besides, room-temperature ferromagnetism in 3D ZCO has been extensively studied and verified by different groups (Physical review letters 93.17 (2004): 177206; Physical review letters 97.3 (2006): 037203; Advanced materials 18.18 (2006): 2476-2480; Nature nanotechnology 4.8 (2009): 523-527). In the revised manuscript, we have modified the 3D BMP model to a 2D version to qualitatively interpret the ferromagnetic couplings. The formation of magnetic polarons owing to the oxygen vacancies gives rise to the ferromagnetic alignment of doped Co atoms. In that sense, the oxygen vacancy also plays a crucial role in the ferromagnetic ground states. The 2008 JAP paper reported the synthesis of ZCO thin films at atmospheric pressure, which might be challenging to form sufficient oxygen vacancies. Deficient oxygen vacancies may fail to percolate and may only provide a weak correlation between the neighboring Co atoms, thus a paramagnetic phase can be more energetically favored under certain circumstances (Nature materials 4.2 (2005):

173-179). We are glad to see that our experimental results are consistent with the BMP theory, which may be helpful in verifying the validity of this model.

The magnetic dipole-dipole interaction is too weak to account for a room-temperature magnetism (three orders of magnitude smaller) (chapter 4.1, Magnetism in Condensed Matter. Stephen Blundell. 238 pp. Oxford U.P., New York, 2001.), and more importantly, the dipolar interaction tends to break a uniform ferromagnetic ordering into domains (Nano letters 18.9 (2018): 5974-5980). In other words, it is something that competes with ferromagnetic ordering rather than something enhancing ferromagnetism. As a consequence, we are confident to exclude the dipole-dipole interaction. On the other hand, the BMP-induced ferromagnetic coupling has been proposed to explain the high- T_c ferromagnetism of DMOs in 3D. Therefore, it is natural for us to resort to the same mechanism in 2D. While the dimensionality effect may result in different parameters in 2D, there hasn't been any report showing that the BMP theory does not apply to the 2D systems. In the revised manuscript, we provide a rough estimate of T_c by modifying the 3D model into a 2D version through a simple dimensionality reduction, which indeed yields a T_c higher than room temperature.

5. Also it is well-known that CoO/Co core can induce exchange bias and thus the strong coercivity, such as JMMM Volume 303, Issue 2, August 2006, Pages e160-e164. Can authors also compare the CoO/Co core-shell hysteresis in Figure 3a to make sure no contribution from the oxidation of Co nanoparticles?

Our response:

The reviewer has pointed out a new idea to exclude CoO nanoparticles in normal systems. Antiferromagnetic CoO can pin the interfacial spins from Co, giving rise to an exchange bias. However, exchange bias may also occur in DMOs without the pinning effect between CoO and Co. In terms of BMP model, the local inhomogeneity of dopant distribution may yield isolated Co atoms or Co-O-Co bonding. Free spins in isolated Co and antiferromagnetic coupling from Co-O-Co are likely to induce a pinning effect on the other ferromagnetically aligned spins. Therefore, a finite exchange bias can be observed in DMOs (Physical Review B 79.11 (2009): 115210), and the exploration of exchange bias may not work to eliminate CoO in 2D ZCO.

Besides, Fig. 1c and Fig. S1 in our manuscript are showing the atomic force microscopy (AFM) results of ZCO atomic layers. The atomically flat surfaces are unambiguously resolved in monolayer and bilayer ZCO without any 3D particles, differing dramatically from the figure 2 in the reference (JMMM Volume 303, Issue 2, August 2006, Pages e160-e164). By combining all the evidences including TEM, AFM, SQUID and XMCD, we are able to exclude the existence of the oxidized Co nanoparticles.

We have cited the paper mentioned by the reviewer in our revised manuscript (ref. 32, paragraph 2, page 4), to strengthen our argument to exclude 3D CoO or Co nanoparticles using AFM.

Reviewer #2 (Remarks to the Author):

The paper presents a wealth of experimental results on an innovative and challenging material. The proposed material is of particular interest due to its new concept that could open new avenues in the search for room temperature ferromagnets suitable for device applications.

The experimental results presented span a wide range of techniques used (XPS, SEM, AFM, TEM, EXAFS, XANES, GIWAXS) to characterize the structural properties of the system, leaving no room, to my opinion, for doubts on the perfectly clean layered structure of the samples.

In addition, the magnetic characterization is also very exhaustive and complete on the basis of MOKE, SQUID and XMCD results, able to rule out contributions from Co clusters and other spurious structures.

Finally, the authors were able to convincingly show the high tunability of the system acting on doping as well as on volume.

Just on this last point, I have a remark for the authors: I think it would be interesting if they could comment a little more on the different samples (#1-#5) shown in Fig. S5 and demonstrating a linear behavior of the saturation magnetization vs volume of the sample: information regarding how the volume is changed (number of layers of gZCO, sample size, planar size..) would be very helpful.

To conclude, I think that the paper is overall very interesting, well written, and discusses original and solid results that could be of wide interest for the community.

Alessandra Continenza

Our response:

We thank the reviewer very much for recommending this manuscript for publication. The linear M_s - V relation indeed sets a good example for the investigations into 2D magnetic nanosheets using SQUID techniques. Alternately stacked ZCO/rGO samples were prepared from one growth session to ensure the consistent Co doping density and oxygen vacancy concentration. Since the thin film thicknesses are also close to each other, we chose five different samples with varying planar sizes. The sample thicknesses were measured by the step profiler and the lateral sizes were determined by a standard scaleplate. Then, the corresponding volumes were calculated, the information of

which is shown in the table below. Through the linear volume dependence of magnetic moments, we can unambiguously eliminate the magnetic signals from random contaminations.

Sample #	Thickness (μm)	Area (mm^2)	Volume (10^{-4} cm^3)
1	21.720	2.25	0.489
2	21.605	4.56	0.985
3	21.770	6.80	1.480
4	21.662	10.80	2.339
5	22.300	15.00	3.345

Reviewer #3 (Remarks to the Author):

Two-dimensional (2D) van der Waals (vdW) magnetic materials are of current great interest for their promising spintronic applications. However, the ferromagnetic ordering temperature is far below room temperature (RT) for those insulating or semiconducting systems. This question is now answered by the present work, which reports tunable RT ferromagnetism in Co-doped 2D vdW ZnO. This work uses transmission electron microscopy and atomic force microscopy results to exclude the existence of metallic Co or cobalt oxides clusters in the Co-doped 2D graphene-like ZnO (gZCO). The authors have observed a spontaneous magnetization in gZCO at room temperature and above, through the magneto-optic Kerr effect, superconducting quantum interference device and X-ray magnetic circular dichroism measurements. The x-ray characterizations suggest that the substitutional Co atoms are in the +2 charge state. Then the authors assume that the substitutional Co^{2+} ions form the bound magnetic polarons and are responsible for the RT ferromagnetism via the impurity-band-exchange interactions.

This is an interesting work and presents important experimental results. Before it can be further considered for publication in Nature Communications, the following points may be addressed by the authors.

Our response:

We appreciate the reviewer for acknowledging the importance of our work. Please find our detailed response to your questions as follows.

1. How can the interstitial Co atoms be excluded? How about the oxygen stoichiometry? Are there oxygen vacancies to introduce the carriers? Otherwise, how would the doped Co^{2+} ions be strongly coupled to form RT ferromagnetism?

Our response:

We truly appreciate the reviewer for the insightful questions. In order to eliminate the interstitial Co atoms, first of all, it is necessary to determine the electronic state of Co ions (Physical review letters 96.2 (2006): 027202). In the manuscript and supplementary information, both XANES (Fig. 2a) and XPS (Fig. S3) indicate a high-spin Co^{2+} state of Co in the 2D ZnO host. Considering that interstitial Co atoms are predicted to take the low-spin state in diluted magnetic oxides (Physical review letters 96.2 (2006): 027202; Physical Review B 68.12 (2003): 125203), the observation of a high-spin Co^{2+} state unveils that the Co atoms are predominantly substituting on the Zn site. Also, the interstitial Co atoms in the DMO are expected to destroy the spin polarization of the surrounding substitutional Co, and hence reduce the average magnetic moments of DMO (Physical review letters 96.2 (2006): 027202; Physical Review B 68.12 (2003): 125203). Such suppressed magnetism is opposite to the strong room-temperature ferromagnetism we have observed in 2D ZCO. Second, the atomic coordination shells of Co are clearly resolved by means of EXAFS (Fig. 2c), which are consistent with the previously reported substitutional Co atoms (Applied Physics Letters 90.10 (2007): 102108). Fine features corresponding to the interstitials are expected between the main peaks (coordination shell) of EXAFS, which are not observed in 2D ZCO. Third, first-principles calculations have suggested that, in 2D graphitic ZnO, Co atoms are energetically barrierless substituted in the Zn sites with energy gain of -2.55 eV (Physical Review B 81.19 (2010): 195413). This implies that substitutional Co atoms occupying Zn sites are more thermodynamically favored than other possibilities, such as interstitials, metallic clusters, and so on. Therefore, we can exclude the interstitial Co atoms.

The investigation into oxygen vacancy along with the BMP theory is indeed an important research direction. One of the major challenges of these 2D magnetic films is the ultra-small mass load due to their 2D nature, posing big challenges for conventional experimental approaches. Considering these challenges in our next steps, we have already planned more in-depth studies by means of SP-STM, in order to image oxygen vacancies, local density of states (carriers) and spin orientations with atomic resolution.

The BMP model was proposed to explain room-temperature ferromagnetism in 3D DMOs, so it is natural for us to resort to the same model in 2D, especially when other commonly seen ferromagnetic exchange (*e.g.* direct exchange, RKKY, etc.) do not apply. When the magnetic polarons originated from oxygen vacancies percolate, they will align the spins of dopants (Co ions) because of the strong s-d exchange coupling, thus yielding a robust long-range ferromagnetic order. However, direct experimental verification of such theory in the 3D bulk crystal is challenging. Now we synthesized 2D version of DMO, which provides a completely new platform that may be utilized to verify the BMP model. We hope our platform will inspire future experiments to confirm the validity of BMP hypothesis.

2. The doped Co^{2+} ions are most likely in the high-spin state due to the low coordination and to the weak crystal field. They have each the spin moment of $3\mu_B$. As the present gZCO is claimed to be a RT ferromagnet, why does the magnetization value of 0.68-0.93 μ_B/Co become much smaller?

Our response:

We thank the reviewer for the nice concern about magnetization values. This question can be addressed as follows. On one hand, the magnetization value is determined by not only the spin moments of individual Co atoms but also their interactive couplings. BMP was proposed to ferromagnetically align these spin moments. However, some factors may lead to a deviation from the perfect ferromagnetic ordering, including but not limited to, non-ideal Co doping and oxygen vacancy concentration, isolated Co ions, misalignment between Co minority band and impurity band, superexchange interaction and thermal fluctuations (Nature materials 4.2 (2005): 173-179). For example, the efficiency of exchange interaction heavily relies on the overlap of electron wavefunction between the minority band of Co and the impurity band formed by Oxygen vacancies (Nature materials 4.2 (2005): 173-179; Physical review letters 93.17 (2004): 177206). Thus, the magnetization value per Co can be smaller than its ideal spin moment, especially at room temperature. On the other hand, our estimated magnetization value is comparable to the previously reported ones (about $0.07 - 2 \mu_B/\text{Co}$) in the 3D ZCO system (Nature materials 4.2 (2005): 173-179), which suggests a general magnetization range of the DMO systems as well as a similar physical mechanism of the magnetic couplings between 2D and 3D DMOs.

3. Besides a set of experimental results showing the RT ferromagnetism, a theoretical input may be expected. The authors have briefly mentioned the picture of bound magnetic polarons and impurity-band-exchange, but the present discussion is still far from being clear. How the doped magnetic impurities form a RT ferromagnetic order in a wide gap semiconductor seems to be a standing issue. The readers may also like to see more theoretical input in the present work.

Our response:

We thank the reviewer's encouragement on the theoretical discussions. We have included more theoretical analyses in the revised manuscript, which are quoted as follows. Due to the scope of this paper, these discussions are preliminary and we hope that our efforts may provoke more in-depth theoretical analyses by colleagues in this field.

(a) Paragraph 1, page 6

BMP was proposed to ferromagnetically align these spin moments. However, some factors may lead to a deviation from the ideal magnetization value, such as non-ideal Co doping and oxygen

vacancy concentration, isolated Co ions, misalignment between Co minority band and impurity band, superexchange interaction, and so on. The magnetization values obtained above are comparable to the previously reported ones (about $0.07 - 2 \mu\text{B}/\text{Co}$) in the 3D ZCO system¹², which suggests a general magnetization range of the DMO systems as well as a similar physical mechanism of the magnetic couplings between 2D and 3D DMOs.

(b) Paragraph 1, page 7

Another important factor is the concentration of Co. We assume one polaron site can ferromagnetically interact with spins within a range of about 10 atom sites considering a hydrogenic orbital of localized carriers¹². Therefore, we expect that a long-range ferromagnetic order will occur at the Co doping level around (percolation threshold)/10. In 2D case, the percolation threshold for the triangular lattice is 50%³⁸, which indicates a requirement of 5% Co to achieve ferromagnetism, and the ferromagnetism will be further enhanced with higher Co occupancy.

(c) Paragraph 1, page 7

More specifically, the cation percolation threshold is 50% in a 2D triangular lattice³⁸, which is much higher than 18% in 3D wurtzite ZnO¹². Therefore, a higher doping concentration of Co is required in 2D ZCO to frustrate the long-range ferromagnetic order, which yields the occurrence of optimal ferromagnetism in a higher doping region than 3D (less than 10%)^{11,12}. On the other hand, by applying the 3D BMP-induced ferromagnetism¹² to 2D through a straightforward dimensionality reduction, we also obtain a rough estimation of T_c of 2D ZCO, which is comparable to or even higher than its 3D counterparts (See Methods). Our experimental observation of room-temperature ferromagnetism and magnetic phase transition is consistent with the BMP theory, which may be helpful in verifying the validity of this model. More in-depth theoretical studies in the future will be helpful to better understand the change of Curie temperature and BMP hypothesis at the 2D limit.

(d) Paragraph 2, page 18

Theoretical calculation

We apply the 3D BMP model¹² to 2D through a straightforward dimensionality reduction (*e.g.* the volume $4\pi r^3/3$ becomes the area πr^2). We obtain the Curie temperature as

$$T_c = \frac{\pi n_o r_c^2 J_{sd}}{2k_B} \sqrt{\frac{S(S+1)}{3}} x\delta,$$

where $n_o = 1.09 \times 10^{19} \text{m}^{-2}$ is the areal density of oxygen atoms (volume density $n_o = 6 \times 10^{28} \text{m}^{-3}$ for the 3D case), r_c is the cation radius (0.06 nm), J_{sd} is the s-d exchange interaction

between impurity-band carriers and localized Co^{2+} ions, $\delta = n_{\text{vac}}/n_O$ is the percentage of oxygen vacancies per unit cell, S is the spin of Co^{2+} ions, and k_B is the Boltzmann constant. If we only take into consideration the geometry change, we can roughly compare the T_c between 2D and 3D cases: $\frac{T_{c,2D}}{T_{c,3D}} = \frac{3}{4} \left(\frac{n_{0,2D}}{n_{0,3D}} \right) \frac{1}{r_c} > 1$. This indicates that the Curie temperature of 2D ZCO can be at least comparable to its 3D counterparts. Other parameters in the model may also be affected by the dimensionality. More in-depth theoretical calculations in the future will provide a better understanding of the Curie temperature and BMP hypothesis at the 2D limit.

Reviewers' Comments:

Reviewer #1:

Remarks to the Author:

Authors took some effort to revise their article and most questions are explained reasonable well. However, regarding the authors' response of question 1, the expression of Curie temperature of 2D case is obtained by straightforward dimensionality reduction of the result of 3D bound magnetic polaron (BMP) model (Nature materials 4. 2 (2005): 173-179), which only takes account the geometrical changes of the overlap between the wavefunction of donor electron and the magnetic cations. If the donor electrons in 2D are not delocalized and the localization length of the donor electrons is not too small, this simple derivation is not unreasonable. However, according to the discussion of BMP theory (Nature materials 4. 2 (2005): 173-179), the hybridization between the donor impurity band and the 3d states of magnetic cation at the Fermi level is necessary for producing the larger exchange energy and high Curie temperature observed in experiment. In 3D BMP model, the cation radius r_c in the expression of Curie temperature is replaced by r_c^{eff} which takes account the hybridization and rescales the exchange energy. Therefore, the statement of that the Curie temperature of 2D ZCO can be comparable to or even higher than its 3D counterparts based on the estimation $T_{(c,2D)}/T_{(c,3D)} = 3/4 (n_{(o,2D)}/n_{(o,3D)}) 1/r_c > 1$ is only valid if the strength of the hybridization between the donor states and Co 3d states does not change upon dimension reduction. I am not sure if the author implied this idea by mentioning "other parameters in the model may also be affected by dimensionality". If this is the case, the author should clarify the meaning of J_{sd} in the expression of Curie temperature. Furthermore, I still think the author should discuss more about whether the dimensionality will affect the hybridization between the donor states and Co 3d states.

Reviewer #3:

Remarks to the Author:

I find the authors have fully addressed all the concerns raised by the three referees. Their reply is satisfactory and the manuscript has been improved, accordingly. Therefore, I recommend its publication in Nature Communications.

Reviewer #1 (Remarks to the Author):

Authors took some effort to revise their article and most questions are explained reasonable well. However, regarding the authors' response of question 1, the expression of Curie temperature of 2D case is obtained by straightforward dimensionality reduction of the result of 3D bound magnetic polaron (BMP) model (Nature materials 4. 2 (2005): 173-179), which only takes account the geometrical changes of the overlap between the wavefunction of donor electron and the magnetic cations. If the donor electrons in 2D are not delocalized and the localization length of the donor electrons is not too small, this simple derivation is not unreasonable. However, according to the discussion of BMP theory (Nature materials 4. 2 (2005): 173-179), the hybridization between the donor impurity band and the 3d states of magnetic cation at the Fermi level is necessary for producing the larger exchange energy and high Curie temperature observed in experiment. In 3D BMP model, the cation radius r_c in the expression of Curie temperature is replaced by r_c^{eff} which takes account the hybridization and rescales the exchange energy. Therefore, the statement of that the Curie temperature of 2D ZCO can be comparable to or even higher than its 3D counterparts based on the estimation $T_{(c,2D)}/T_{(c,3D)} = 3/4 (n_{(o,2D)}/n_{(o,3D)}) 1/r_c > 1$ is only valid if the strength of the hybridization between the donor states and Co 3d states does not change upon dimension reduction. I am not sure if the author implied this idea by mentioning "other parameters in the model may also be affected by dimensionality". If this is the case, the author should clarify the meaning of J_{sd} in the expression of Curie temperature. Furthermore, I still think the author should discuss more about whether the dimensionality will affect the hybridization between the donor states and Co 3d states.

Our response:

We really appreciate the reviewer for pointing out another important factor to affect the Curie temperature, which is the hybridization between donor impurity band and Co 3d states. The strength of hybridization will leave an impact on the s-d exchange parameter J_{sd} and the effective cation radius. We fully agree that the dimensionality will also affect the Curie temperature through its impact on the hybridization. We indeed implied factors like the hybridization by mentioning 'other parameters in the model may also be affected by dimensionality'. From 3D to 2D, quantum confinement and structural phase transition will yield a variation of the wavefunction overlap between donor electrons and doped Co^{2+} ions. Our experimental data have indicated that the Curie temperature and magnetization value (per Co atom) in 2D is comparable to or larger than their 3D counterparts. Further in-depth theoretical studies of s-d hybridization will be needed in the future to help the community better understand the 2D BMP theory from electronic structures. We are happy that our material system offers such a platform for future explorations in this direction.

In order to avoid the confusion, we are including the hybridization factor in our discussion, 'Other parameters in the model may also be affected by the dimensionality. For instance, from 3D to 2D, the quantum confinement and structural phase transition can induce a variation of hybridization between donor impurity band and Co 3d orbitals, which will affect the strength of s-d exchange parameter J_{sd} , and thus the Curie temperature. More in-depth theoretical calculations in the future will provide a better understanding of the Curie temperature and BMP hypothesis at the 2D limit' (Paragraph 2, page 16)

Reviewer #3 (Remarks to the Author):

I find the authors have fully addressed all the concerns raised by the three referees. Their reply is satisfactory and the manuscript has been improved, accordingly. Therefore, I recommend its publication in Nature Communications.

Our response:

We thank the reviewer very much for helping improve our work and for recommending our revised manuscript to publication.